# Long-term elevated levels of loneliness are linked to lower health-related quality of life in middle-aged Australian women
Neta HaGani[1,2], Katherine Owen[1,2], Philip J. Clare [1,2,3], Dafna Merom [4], Ben J. Smith[1,2] & Ding Ding [1,2] ✉

Loneliness has long been associated with poor health outcomes. However, few studies have considered the dynamic nature of loneliness over time. This study aimed to identify longitudinal patterns of loneliness over 18 years and their associations with physical and mental health-related quality of life. Using data from the Australian Longitudinal Study of Women's Health, we conducted a Latent Class Analysis to identify long-term loneliness patterns. We used Multinomial logistic regression to examine baseline predictors of loneliness trajectories and linear regression to examine the association between loneliness trajectories and health-related quality of life. Baseline predictors, such as smoking, depression, anxiety, stress and low social support, were associated with higher odds of 'Increasing', 'Stable-medium' and 'Stable-high loneliness. Compared to 'Stable-low loneliness, 'Increasing' [B = −3.73 (95%CI = −5.42, −2.04)], 'Medium' [B = −3.12 (95%CI = −5.08, −1.15)] and 'High' loneliness [B = −5.67 (95%CI = −6.84, −4.49)] were associated with lower mental health-related quality of life. 'Increasing' loneliness was also associated with lower physical health-related quality of life [B = −1.06 (95%CI = −2.11, −0.02)]. Among health-related quality of life sub-scales, emotional role, social functioning and physical role were the most strongly associated with loneliness. Findings highlight the importance of addressing loneliness among women to promote their health and well-being.

Loneliness stems from a perceived deficit in social relationships, often described as a gap between the desired and actual levels of social connections[1]. While episodic loneliness is a common human experience and considered an important survival mechanism linked to pain and hunger[2], chronic loneliness has been associated with various physical and mental illnesses such as diabetes, heart disease, cancer, low social functioning and premature death[3–8]. Loneliness has also been associated with risky health behaviours such as smoking[9,10] and with a higher likelihood of risky drinking[11].

There is also evidence of a strong inverse association between loneliness and health-related quality of life[12,13]. Health-related quality of life is a multidimensional concept that goes beyond the presence or absence of disease. It encompasses diverse aspects of physical, mental and social health and functioning[14]. Health-related quality of life is a common measure in medicine and public health for assessing and predicting population health outcomes and mortality[15–17].

The association between loneliness and health-related quality of life has been demonstrated across different populations, such as older adults and patients with multimorbidity[18–20]. While loneliness can contribute to worsening health outcomes[21], poor health can also lead to increased loneliness due to challenges in fulfilling different social roles[22,23]. To date, most studies that examined the association between loneliness and health-related quality of life were cross-sectional[3], and the small number of longitudinal studies only assessed loneliness at one point in time, which was unable to capture the long-term patterns of loneliness across the lifespan[13,24–26].

Loneliness is a dynamic experience that evolves throughout life. During middle-to-older age significant life transitions, such as retirement, empty nesting and the loss of loved ones[27,28], as well as an unhealthy lifestyle[29] can disrupt established routines and social networks, thus triggering a sense of loneliness and decreased health-related quality of life[30,31]. Moreover, the middle-to-older stage of life often coincides with a higher likelihood of developing chronic illnesses and disabilities, further contributing to

[1]Prevention Research Collaboration, Sydney School of Public Health, The University of Sydney, Sydney, NSW, Australia. [2]Charles Perkins Centre, The University of Sydney, Sydney, NSW, Australia. [3]National Drug and Alcohol Research Centre, UNSW Sydney, Sydney, NSW, Australia. [4]School of Health Science, Western Sydney University, Sydney, NSW, Australia. ✉e-mail: melody.ding@sydney.edu.au

loneliness and health-related quality of life[32]. As a result, addressing the dynamic nature of loneliness is critical to understanding and enhancing health-related quality of life for individuals in mid-to-older age.

To better understand the long-term experience of loneliness among middle-aged and older adults and inform interventions to improve health-related quality of life across the lifespan, it is essential to provide an accurate depiction of the association between long-term trajectories of loneliness and the various dimensions of health-related quality of life. Therefore, the aims of the current study are threefold: 1) to identify the patterns of loneliness spanning from middle-to-older age; 2) to identify the demographic characteristics and health behaviours associated with different loneliness patterns and 3) to examine the association between loneliness trajectories and physical and mental dimensions of health-related quality of life.

## Methods

The current study followed the strengthening of the reporting of observational studies in epidemiology (STROBE) guidelines[33] (Table S1.1). As this was an exploratory analysis and no hypotheses were made, a pre-registration protocol was not required.

### Study population

Data were derived from nine waves of the Australian Longitudinal Study of Women's Health (ALSWH) 1946–51 population-representative cohort[34]. Participants were between the ages of 43 and 51 at baseline in the year 1996 and were originally selected from the Medicare database to represent Australian citizens and permanent residents. Sampling from the population followed a random selection process within each age group, with the exception that women residing in rural and remote areas were sampled at a rate doubling that of women in urban areas. This approach ensured that the number of women living outside major urban areas was sufficient for statistical comparisons between the circumstances and health of city and rural women, a significant issue for Australia both now and in the future.

The cohort included 13,714 women at baseline. Collected at 3-year intervals starting from 1996 (baseline), the data included self-reported information on health status, health behaviours, physical function, and social and mental well-being. Self-completed questionnaires were available online or in paper form via mail. A free call number was provided for additional assistance with the survey. All telephone surveys were conducted by experienced interviewers and qualified health services interpreters. Confidentiality was maintained by storing all data under identification numbers, with names and addresses kept separate from the main data files. All staff were required to comply with the provisions of the Privacy Act. The response rate was between 53% and 56%[35]. Ethical approval for this study was obtained from the Human Research Ethics Committees (HRECs) of the Universities of Newcastle and Queensland (approval numbers H-076-0795 and 2004000224). All participants provided their informed consent prior to their participation[36] and more information about the study can be found elsewhere[37].

### Measures

**Main exposure: loneliness.** Loneliness was assessed in waves 3 (year 2001, participants aged 48–55 years) to wave 8 (year 2016, aged 64–71 years) using a single item from the Centre for Epidemiologic Studies Depression Scale (CES-D)[38]. This item measured the extent of loneliness experienced in the past week on a 4-point Likert scale: 1 = Rarely or none of the time (less than 1 day); 2 = Some or a little of the time (1–2 days); 3 = Occasionally or a moderate amount of time (3–4 days); and 4 = Most or all of the time (5–7 days). These responses were recategorised into a binary variable: 0 for levels 1–2 in the original scale and 1 for levels 3–4 in the original scale. One-item loneliness measures were previously shown to be reliable ($r_{xx} > 0.70$[39]) and valid (strongly correlated with other more comprehensive, common loneliness measures such as the UCLA and the de Jong loneliness scales, as well as with the indirect one-item loneliness measures[39–41]).

**Main outcome: health-related quality of life.** The health-related quality of life of participants was assessed using the SF-36, a self-reported questionnaire that encompasses eight dimensions of health and well-being in the past 4 weeks or on a typical day, namely[42]: 1) vitality, the amount of energy or fatigue; 2) physical functioning, the ability to perform different types of vigorous or moderate activities on a typical day; 3) bodily pain, which is the severity of pain experienced in general and at work; 4) general health ratings overall and in comparison to others; 5) role limitations due to physical health, including difficulties performing activities or accomplishing work due to physical limitations; 6) role limitations of the same types due to emotional problems; 7) social role functioning, the extent to which physical health or emotional problems interfered with normal social activities with family, friends, neighbours, or groups; and, 8) mental health/emotional well-being, including feeling nervous, peaceful, happy etc. Each dimension is quantified on a scale from 0 to 100, where higher values reflect a more positive perception of health. The individual sub-scales were subsequently aggregated to create two overarching summary measures: 1) Physical Component Summary (PCS), which includes the sub-scales for general health, physical functioning, role physical, and bodily pain, and 2) Mental Component Summary (MCS), that combines the sub-scales for vitality, social functioning, role emotional, and mental health. The SF-36 is a validated and reliable measure that was examined among a variety of populations, including middle-aged and older adults[43,44]. In a population-representative Australian household survey (Australian National Household Survey), the internal consistency ranged between 0.82-0.93 across the health-related quality of life dimension[43]. Similar reliability was observed in other studies among diverse populations[45–47].

### Baseline predictors of loneliness

Information regarding the re-categorisation of the baseline predictors is provided in the Supplementary Notes. To ensure that the baseline predictors occurred before the outcome (loneliness), they were derived from waves 1–2 and included: age; marital status (partnered, not partnered, widowed); current employment status (no, yes); living arrangement (live alone/ live with others); highest level of education completed (high school or less, tertiary not university, college/university); language spoken at home (English, non-English European, Asian, other); country of birth (Australia, other); area-level socioeconomic status using the Index of Relative Socio-Economic Disadvantage (IRSD)[48]; location of residence based on the postcode-level Accessibility-Remoteness Index of Australia Plus (ARIA+; major city, regional, remote)[49]; self-reported body mass index (BMI; underweight, healthy, overweight, obese); smoking status (never smoked, ex-smoker, currently smoking); self-reported previous diagnosis or symptoms of anxiety and depression; current depression based on the Centre for Epidemiological Studies-Depression (CES-D) scale[38]; risky alcohol consumption (consuming more than 10 standard drinks in a week or more than four standard drinks per day) and binge drinking (consuming five or more drinks on a single occasion) based on the 2020 National Health Medical Research Council guidelines[50]; social support using the Medical Outcomes Study (MOS) measure of the frequency of receiving support from others on a scale of 1–5 (higher score reflects better support); stress measured using the Perceived Stress Questionnaire (mean score of 10 items ranging from 0 to 4)[51].

### Data analysis

Initially, we constructed a range of latent class models encompassing one to five loneliness classes. Goodness-of-fit through metrics such as Akaike's Information Criterion (AIC), Bayesian Information Criterion (BIC), and Sample-Size Adjusted BIC (SSABIC) was used to identify the most suitable number of classes. Because the sample was not random, the latent class analysis was estimated using sample weights. Next, multivariable multinomial logistic regression models were employed to examine the relationship between demographic characteristics and loneliness patterns, using multinomial odds ratios (mOR) as effect sizes.

Next, linear regression models were employed to investigate the association between loneliness class membership from Waves 3 to 8 and health-related quality of life at Wave 9 as a continuous outcome. The assumptions of linear regression models were tested. The data appear to show slight non-normality, especially in the tails, which is typical with very large datasets[52] (Fig. S3.1). The results of the linear regressions are presented as unstandardised coefficients. Since individuals cannot be definitively assigned to latent classes, the three-step Bolck–Croon–Hagenaars estimation method (BCH) was employed to accommodate the uncertainty in classification[53]. All models were weighted using the BCH weight, and distal outcome models adjusted for variables including age, education, marital status, country of birth, language spoken at home, living arrangement, area remoteness, area-level socioeconomic status, employment, smoking, alcohol consumption, anxiety, depression, stress, BMI, social support and baseline health-related quality of life. These analyses were conducted using Stata Version 17.0[54].

To assess the robustness of the results and to examine the confounding effect of baseline predictors on the association between loneliness and health-related quality of life, the linear regression models were run 1) without adjustments for any baseline predictors and 2) adjusting for sociodemographic baseline predictors only (i.e. without the mental health, social support and health behaviour variables).

**Missing data.** There were 9154 (66.75%) incomplete cases, which contributed to a total of 20.7% missing data in the dataset. Figures S3.2 and S3.3 present the missing data for each variable used in the model. Little's test indicated that data were not missing completely at random and therefore it was assumed to be missing at random. Multiple imputations were used to handle missing data, with twenty imputations using the Mice package in R version 4.3.1[55], with random forests to accommodate potential non-linear relationships and interactions during the data imputation process[56]. Analyses were conducted on all twenty datasets (imputations), and the resulting estimates were aggregated using Rubin's rules[57].

**Sensitivity analysis.** Further assessments were conducted to test the robustness of the models. We conducted the latent class models using a Probit link function, which assumes that the error terms follow a standard normal distribution, compared to the logit model, which assumes the error terms follow a logistic distribution. We also conducted a sensitivity analysis using a different cut point of the outcome variable loneliness (1. rarely or none of the time vs. levels 2–4). The reversed cut point (the lowest loneliness levels 1–3 vs. the highest level 4) was not tested as the highest loneliness category (most or all of the time) constituted less than 5% of the sample.

## Reporting summary

Further information on research design is available in the Nature Portfolio Reporting Summary linked to this article.

## Results

### Demographic characteristics

Table 1 shows the sociodemographic characteristics of the sample at baseline (n = 13,714). The flow diagram of participants included in the study can be found in Table S3.1. The median age was 49.5 years (interquartile range = 2.5), and the majority of participants were married or partnered (83.9%), had a high school education (67.1%) and were employed (79.5%). Most participants were Australian-born (76.3%) and spoke English at home (94.0%), and 9.1% reported experiencing loneliness 3–7 days a week. The MOS social support score was 3.9 (SD = 1.0, Range = 1–5).

### Loneliness latent classes

Goodness-of-fit as assessed by AIC, BIC and SSABIC suggested the best fit was a four-class model, which indicated optimal model fit (Table S3.2). The average classification probabilities are 0.67 for class 1, 0.91 for class 2, 0.67 for class 3 and 0.77 for class 4. The average classification probabilities refer to

**Table 1 | Baseline characteristics of participants—The Australian Longitudinal Study of Women's Health (ALSWH) 1946–51 cohort (waves 1–2; n = 13,714)**

| | | Median (IQR) | SD |
|---|---|---|---|
| Age (years) | | 49.5 (2.5) | 1.5 |
| | | **n** | **%** |
| Education | High school or less | 9200 | 67.1 |
| | Vocational | 2613 | 19.1 |
| | College/ university | 1901 | 13.9 |
| Marital status | Partnered | 11,509 | 83.9 |
| | Single/divorced/ separated | 1928 | 14.1 |
| | Widowed | 277 | 2.0 |
| Location of residence | Major city | 4604 | 33.6 |
| | Regional | 8441 | 61.6 |
| | Remote | 669 | 4.9 |
| Currently employed | Yes | 10,906 | 79.5 |
| Language spoken at home | English | 12,885 | 94.0 |
| Born in Australia | Yes | 10,461 | 76.3 |
| Living arrangement | Live alone | 759 | 5.5 |
| Smoking status | Never smoked | 8060 | 58.8 |
| | Ex-smoker | 3483 | 25.4 |
| | Currently smoking | 2171 | 15.8 |
| Body Mass Index | Underweight (<18.5) | 175 | 1.3 |
| | Healthy (18.5 ≤ BMI ≤ 24.9) | 6705 | 48.9 |
| | Overweight (25.0 ≤ BMI ≤ 29.9) | 4183 | 30.5 |
| | Obese (≥30.0) | 2651 | 19.3 |
| Depression[a] | Yes | 4130 | 30.1 |
| Anxiety[b] | Yes | 4279 | 31.2 |
| Risky alcohol consumption (>10 drinks p/week)[c] | Yes | 1681 | 12.3 |
| Loneliness (feeling lonely 3+ days in a week) | Yes | 1243 | 9.1 |
| | | **Mean (range)** | **SD** |
| MOS Social support score[d] | | 3.9 (1–5) | 1.0 |
| Stress score[e] | | 0.6 (0–4) | 0.5 |

*MOS* Medical Outcomes Study, *IRQ* Interquartile range.
[a]Depression: ever being told by a doctor about having depression or having symptoms of depression.
[b]Anxiety: ever being told by a doctor about having anxiety or having symptoms of anxiety.
[c]Risky alcohol consumption: consuming more than 10 standard drinks in a week or more than four standard drinks per day.
[d]Higher scores represent better outcomes.
[e]Higher scores represent worse outcomes.

the average likelihood that a given observation belongs to each of the respective classes. For example, for Class 1, the average classification probability is 0.67, indicating that on average, the model assigns 67% likelihood to observations being classified as Class 1 (Fig. 1).

Class 1 was labelled as 'Increasing' loneliness (n = 10,121, 73.8%) due to the increasing trend of loneliness throughout time. The mean loneliness across waves 3–8 among individuals in Class 1 was 33.5%. Class 2 was labelled as 'Stable-low' loneliness (n = 727, 5.3%), which was characterised by consistently low loneliness across all waves (Mean $_{(waves\ 3–8)}$ = 2.8%). Class 3 was labelled as 'Stable-medium' loneliness (n = 2345, 17.1%) and was characterised by a stable trend of medium-level loneliness throughout time (Mean $_{(waves\ 3–8)}$ = 38.1%), and class 4 was labelled as 'Stable-high' loneliness

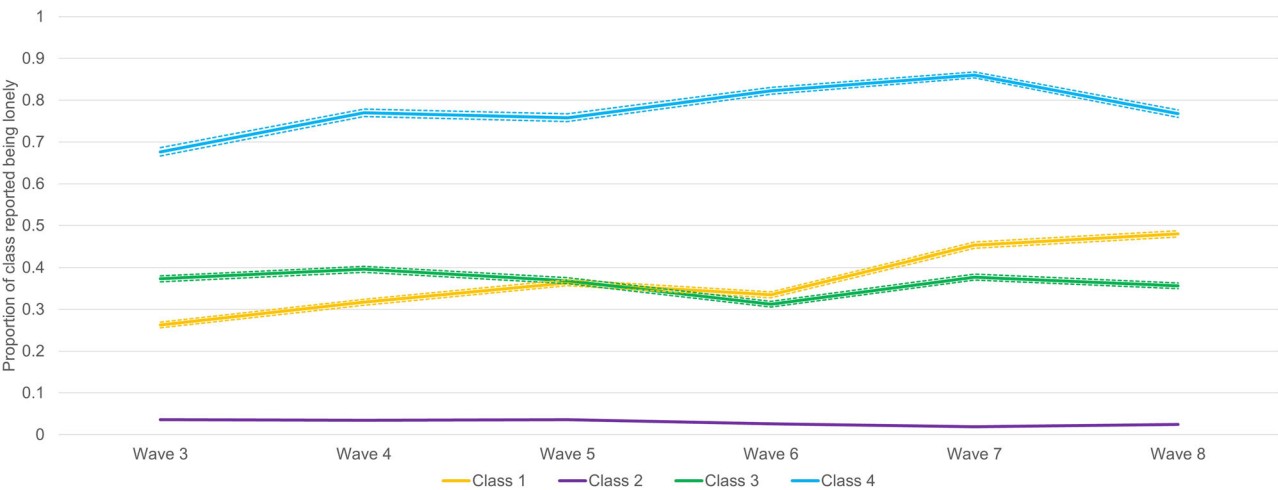

**Fig. 1 | Probabilities of loneliness for each loneliness class over time (n = 13,714).** Class 1 = 'Increasing' loneliness; Class 2 = 'Stable-low' loneliness; Class 3 = 'Stable-medium' loneliness; Class 4 = 'Stable-high' loneliness. 95% Confidence intervals.

(n = 507, 3.8%) and was characterised by a stable high level of loneliness throughout time (Mean $_{(waves\ 3-8)}$ = 77.2%).

**Predictors of loneliness latent class**
Table 2 presents the predictors of the latent class membership with 'Stable-low' loneliness as the comparison class. Widows had higher odds of being in the 'Stable-high' loneliness class [mOR = 2.08 (95%CI = 1.15, 3.78)]. Those who were current smokers had higher odds of being in the 'Increasing' loneliness [mOR = 1.24 (95%CI = 1.01, 1.54)] or 'Stable-medium' [mOR = 1.28 (95%CI = 1.01, 1.61)] classes than non-smokers. A higher CES-D score was associated with higher odds of being in the 'Increasing loneliness' [mOR = 1.06 (95%CI = 1.02, 1.09)], 'Stable-medium' [mOR = 1.07 (95%CI = 1.04, 1.11)] or 'Stable-high' [mOR = 1.11 (95%CI = 1.07, 1.15)] loneliness classes. A higher stress score was associated with higher odds of being in the 'Increasing' loneliness [mOR = 1.37 (95%CI = 1.12, 1.68)], 'Stable-medium' [mOR = 1.36 (95%CI = 1.09, 1.70)] or 'Stable-high' [mOR = 1.72 (95%CI = 1.34, 2.20)] classes. Reporting a past depression diagnosis was associated with higher odds of being in the 'Stable-medium' loneliness class [mOR = 1.47 (95%CI = 1.11, 1.93)]. Lower social support was associated with higher odds of being in the 'Increasing' loneliness [mOR = 0.75 (95%CI = 0.62, 0.90)], 'Stable-medium' [mOR = 0.70 (95%CI = 0.59, 0.84)] or 'Stable-high' [mOR = 0.55 (95%CI = 0.48, 0.62)] classes.

**Health-related quality of life predicted by loneliness class**
Multivariate linear regression was conducted to examine the unadjusted, sociodemographic-adjusted (without health, mental health and risky health behaviours) and fully adjusted associations (the main model), between loneliness classes and health-related quality of life components, with 'Stable-low' loneliness as the comparison group. The results are presented as unstandardised coefficients (Tables 3 and S3.3).

In the fully adjusted models, increasing loneliness compared to 'Stable-low' loneliness was associated with a significantly lower quality of life at wave 9 on all the physical and mental scales and sub-scales. 'Increasing' loneliness was inversely associated with the overall physical component [B = −1.06 (95%CI = −2.11, −0.02)]. The strongest associations with physical components were in the sub-scales of physical role limitations [B = −3.19 (95% CI = −6.27, −0.11)], pain [B = −4.36 (95%CI = −7.05, −1.67)] and general health [B = −4.36 (95%CI = −7.05, −1.67)]. 'Increasing' loneliness had a strong association with the overall mental component [B = −3.73 (95% CI = −5.42, −2.04)]. The strongest associations with mental functioning sub-scales were found in emotional role limitations [B = −9.57 (95%CI = −14.30, −4.84)], social functioning [B = −7.53 (95%CI = −10.67, −4.39)] and mental health [B = −6.50 (95%CI = −9.58, −3.42)].

Compared to 'Stable-low' loneliness, the 'Stable-medium' loneliness class was associated with significantly lower scores on health-related quality of life components and sub-scales, except for the overall physical component summary and the physical functioning sub-scale. However, an inverse association with health-related quality of life was observed for physical role limitations [B = −7.18 (95%CI = −12.60, −1.76)], pain [B = −4.36 (95% CI = −7.05, −1.67)] and general health [B = −4.36 (95%CI = −7.05, −1.67)]. In addition, strong inverse associations were found with the mental component summary scale [B = −3.12 (95%CI = −5.08, −1.15)] and its sub-scales, with the strongest association occurring with emotional role limitations [B = −8.12 (95%CI = −13.44, −2.80)] and social functioning [B = −6.38 (95%CI = −10.07, −2.69)].

Compared to 'Stable-low' loneliness, the 'Stable-high' loneliness class was strongly associated with all of the health-related quality of life components except for the overall physical component summary. The strongest associations with physical sub-scales were physical role limitations [B = −9.69 (95%CI = −13.72, −5.66)] and general health perceptions [B = −4.78 (95%CI = −7.00, −2.55)]. 'Stable-high' loneliness was also strongly associated with all of the mental health components, including the overall mental component summary [B = −5.67 (95%CI = −6.84, −4.49)] and its sub-scales. The strongest associations were found with emotional role limitations [B = −15.25 (95%CI = −13.48, −7.88)], social functioning [B = −10.68 (95%CI = −6.84, −4.49)] and emotional well-being [B = −9.30 (95%CI = −11.36, −7.25)]. The unadjusted and sociodemographic-adjusted models followed similar patterns to the main model but were of a larger magnitude.

**Sensitivity analysis**
Results from the probit model were consistent with the main analyses (Tables S4.1 and S4.2). When using the different loneliness cut points, we found that the four-class model was still the optimal fit (Table S4.3) and similar associations were found between the predictors of the latent class membership and the four classes. However, as expected, when defining the loneliness group as those who responded as being lonely "some or a little of the time", the associations became weaker and less precise than those in the main analysis, except in the cases of depression and social support (Table S4.4). Attenuations were also observed in the associations between loneliness classes and health-related quality of life components in the unadjusted, sociodemographic-adjusted and fully adjusted models. There were no significant associations in the 'Increasing' loneliness class compared to the 'Stable low'. However, in the 'Stable medium' and 'Stable high' classes, loneliness was significantly associated with almost all of the health-related quality of life components (Table S4.5). Overall, these sensitivity analyses have demonstrated the robustness of our findings.

**Table 2 | Multivariate multinomial logistic regression predicting latent class membership using baseline predictors (n = 13,714)**

| | Increasing[a] mOR (95%CI) | Stable-medium[a] mOR (95%CI) | Stable-high[a] mOR (95%CI) |
|---|---|---|---|
| Age | 1.00 (0.95, 1.06) | 1.01 (0.95, 1.07) | 1.01 (0.93, 1.09) |
| Education | | | |
| High school or less | Reference | Reference | Reference |
| Tertiary not university | 0.90 (0.73, 1.11) | 0.90 (0.71, 1.15) | 0.82 (0.57, 1.19) |
| College/ university | 0.96 (0.74, 1.26) | 0.95 (0.72, 1.27) | 0.76 (0.48, 1.19) |
| Living arrangement | | | |
| Live alone | Reference | Reference | Reference |
| Live with others | 1.23 (0.79, 1.91) | 1.10 (0.63, 1.92) | 0.98 (0.61, 1.58) |
| Marital status | | | |
| Partnered | Reference | Reference | Reference |
| Non-partnered | 1.21 (0.94, 1.54) | 1.28 (0.97, 1.69) | 1.28 (0.92, 1.78) |
| Widowed | 1.56 (0.87, 2.80) | 1.70 (0.98, 2.94) | 2.08 (1.15, 3.78) |
| Currently employed | | | |
| No | Reference | Reference | Reference |
| Yes | 0.95 (0.77, 1.18) | 0.99 (0.79, 1.24) | 0.85 (0.64, 1.12) |
| IRSD | 1.00 (1.00, 1.00) | 1.00 (1.00, 1.00) | 1.00 (1.00, 1.00) |
| Location of residence | | | |
| Major city | Reference | Reference | Reference |
| Regional | 1.04 (0.82, 1.32) | 1.10 (0.84, 1.46) | 0.95 (0.71, 1.26) |
| Remote | 1.01 (0.65, 1.58) | 0.97 (0.57, 1.64) | 1.37 (0.77, 2.42) |
| Born in Australia | | | |
| Yes | Reference | Reference | Reference |
| No | 1.01 (0.77, 1.31) | 0.94 (0.71, 1.25) | 0.86 (0.62, 1.20) |
| Language spoken at home | | | |
| English | Reference | Reference | Reference |
| European | 1.07 (0.65, 1.75) | 1.20 (0.68, 2.12) | 1.14 (0.60, 2.15) |
| Asian | 0.89 (0.32, 2.48) | 0.96 (0.31, 2.92) | 1.08 (0.30, 3.81) |
| Other | 1.06 (0.38, 2.98) | 1.34 (0.46, 3.87) | 1.73 (0.51, 5.90) |
| Risky alcohol consumption (>10 drinks p/week)[b] | | | |
| No | Reference | Reference | Reference |
| Yes | 1.08 (0.74, 1.57) | 1.24 (0.87, 1.79) | 0.98 (0.64, 1.50) |
| Binge drinking (>1 episode p/week) | | | |
| No | Reference | Reference | Reference |
| Yes | 1.10 (0.85, 1.43) | 1.00 (0.76, 1.33) | 0.98 (0.72, 1.35) |
| Smoking status | | | |
| Non-smoker | Reference | Reference | Reference |
| Ex-smoker | 1.09 (0.86, 1.36) | 1.06 (0.82, 1.37) | 1.05 (0.75, 1.45) |
| Current smoker | 1.24 (1.01, 1.54) | 1.28 (1.01, 1.61) | 1.13 (0.80, 1.61) |
| Body mass index | | | |
| Healthy (18.5 ≤ BMI ≤ 24.9) | Reference | Reference | Reference |
| Underweight (<18.5) | 0.86 (0.41, 1.80) | 0.86 (0.40, 1.85) | 1.41 (0.60, 3.32) |
| Overweight (25.0 ≤ BMI ≤ 29.9) | 1.12 (0.91, 1.37) | 1.05 (0.83, 1.33) | 1.28 (0.94, 1.73) |
| Obese (≥30.0) | 1.18 (0.93, 1.49) | 1.08 (0.81, 1.45) | 1.55 (1.12, 2.15) |
| Depression (CES-D score) | 1.06 (1.02, 1.09) | 1.07 (1.04, 1.11) | 1.11 (1.07, 1.15) |
| Stress score | 1.37 (1.12, 1.68) | 1.36 (1.09, 1.70) | 1.72 (1.34, 2.20) |

**Table 2 (continued) | Multivariate multinomial logistic regression predicting latent class membership using baseline predictors (n = 13,714)**

| | Increasing[a] mOR (95%CI) | Stable-medium[a] mOR (95%CI) | Stable-high[a] mOR (95%CI) |
|---|---|---|---|
| Depression[c] | | | |
| No | Reference | Reference | Reference |
| Yes | 1.34 (0.98, 1.83) | 1.47 (1.11, 1.93) | 1.34 (0.97, 1.85) |
| Anxiety[d] | | | |
| No | Reference | Reference | Reference |
| Yes | 1.06 (-0.16, 0.27) | 1.00 (0.78, 1.28) | 0.83 (0.60, 1.15) |
| Social support score | 0.75 (0.62, 0.90) | 0.70 (0.59, 0.84) | 0.55 (0.48, 0.62) |

mOR Multinomial odds ratio, CES-D Center for Epidemiological Studies- Depression, IRSD Index of Relative Socio-Economic Disadvantage.
[a]Reference class: 'Stable-low'. mORs refer to the odds of having the attribute in a certain class compared to the stable-low class.
[b]Risky alcohol consumption: consuming more than 10 standard drinks in a week or more than four standard drinks per day.
[c]Depression: ever being told by a doctor about having depression or having symptoms of depression.
[d]Anxiety: ever being told by a doctor about having anxiety or having symptoms of anxiety.

## Discussion

The current study reveals that over the course of 18 years, elevated levels of loneliness, whether 'Increasing' or stable at a 'Medium' or 'High' level, were linked to a lower overall mental health-related quality of life. Additionally, the 'Increasing' loneliness class was the only class correlated with lower overall physical health-related quality of life. Out of the health-related quality of life sub-scales, emotional role, social functioning, and physical role had the strongest association with loneliness across classes. The current findings strengthen the previous evidence by using a longitudinal design to examine the associations between loneliness patterns and health-related quality of life over the longest period of time to our knowledge (18 years).

Our research sheds light on the dynamic nature of loneliness, revealing variations in its associations with health-related quality of life over time. The 'Stable-high' class was associated with the most adverse outcomes, with a significantly lower health-related quality of life in all scales and sub-scales, except for the overall physical health measure. 'Stable-high' loneliness was also associated with several risk factors such as obesity, depression, stress and less social support. Previous studies showed that prolonged (chronic) loneliness is a more serious health concern compared to transient loneliness[58]. Our findings from the 'Stable-high' class echo this finding, suggesting that high chronic loneliness remains a major risk to overall health[58,59]. Our findings also reveal that 'Stable-high' loneliness was more likely to occur in widows, which aligns with previous studies showing associations between widowhood and loneliness[60,61]. Therefore, despite the relatively small size of the 'Stable-high' class (4% of the total sample), it represents a high-risk category and therefore should be prioritised for health promotion interventions.

While the 'Stable-medium' loneliness had a weaker association with health-related quality of life compared to the 'Stable-high' and 'Increasing' loneliness classes, it was still inversely associated with physical and emotional role functions. These findings suggest that even a moderate amount of loneliness, over time, may be associated with a major risk for health. Given that this class constituted 17% of the sample, it is important to be vigilant to the potentially deteriorating health-related quality of life of this group. Incorporating loneliness assessments into primary healthcare for population screening targeting moderate loneliness, and facilitating referrals to community programs and interventions, may be important for alleviating the experience of loneliness and improving health-related quality of life[62-64].

The predominant loneliness class (78.3%) identified in this study was those who experienced a progressive increase in loneliness over time. This class initiates with a moderate level of loneliness prevalence, similar to the 'Stable-medium' loneliness, and increases gradually up to an average of 50% loneliness prevalence within this class. This pattern reflects the increase of loneliness from middle-to-older age, which has been found previously among women[65]. 'Increasing' loneliness was the only class that was

**Table 3 | Multivariate multinomial linear regression predicting latent class membership using baseline predictors (Beta coefficients, 95%CI): data fully adjusted, adjusted only for sociodemographic characteristics and unadjusted (n = 13,714)**

| | | Increasing B (95%CI) | Stable-medium B (95%CI) | Stable-high B (95%CI) | M (SD) |
|---|---|---|---|---|---|
| PCS | Fully adjusted | −1.06 (−2.11, −0.02) | −0.83 (−2.01, 0.35) | −0.78 (−1.87, 0.31) | 44.95 (10.48) |
| | Sociodemographic adjusted | −3.65 (−5.10, −2.19) | −3.70 (−4.98, −2.41) | −4.85 (−6.22, −3.49) | |
| | Unadjusted | −4.02 (−5.55, −2.49) | −4.18 (−5.45, −2.90) | −5.58 (−6.99, −4.18) | |
| MCS | Fully adjusted | −3.73 (−5.42, −2.04) | −3.12 (−5.08, −1.15) | −5.67 (−6.84, −4.49) | 52.53 (9.56) |
| | Sociodemographic adjusted | −6.96 (−8.69, −5.23) | −6.99 (−8.67, −5.31) | −10.53 (−12.11, −8.95) | |
| | Unadjusted | −7.19 (−8.98, −5.41) | −7.26 (−8.97, −5.55) | −10.93 (−12.53, −9.33) | |
| Physical functioning | Fully adjusted | −3.19 (−6.27, −0.11) | −2.59 (−5.79, 0.61) | −4.32 (−7.04, −1.60) | 74.66 (23.15) |
| | Sociodemographic adjusted | −9.38 (−13.32, −5.45) | −9.50 (−12.74, −6.27) | −14.00 (−17.18, −10.82) | |
| | Unadjusted | −10.36 (−14.44, −6.27) | −10.73 (−13.88, −7.57) | −15.88 (−19.16, −12.60) | |
| Role physical | Fully adjusted | −8.09 (−12.59, −3.58) | −7.18 (−12.60, −1.76) | −9.69 (−13.72, −5.66) | 71.04 (40.9) |
| | Sociodemographic adjusted | −18.80 (−25.03, −12.58) | −19.51 (−24.71, −14.31) | −26.31 (−31.38, −21.24) | |
| | Unadjusted | −20.06 (−26.52, −13.59) | −21.11 (−26.29, −15.93) | −28.65 (−33.85, −23.45) | |
| Pain index | Fully adjusted | −4.36 (−7.05, −1.67) | −3.45 (−6.51, −0.38) | −4.67 (−6.96, −2.38) | 66.29 (23.83) |
| | Sociodemographic adjusted | −11.05 (−14.20, −7.90) | −11.05 (−14.01, −8.08) | −14.97 (−18.04, −11.90) | |
| | Unadjusted | −11.77 (−15.01, −8.53) | −11.94 (−14.89, −8.99) | −16.33 (−19.51, −13.14) | |
| General health | Fully adjusted | −3.99 (−6.15, −1.84) | −3.29 (−5.75, −0.83) | −4.78 (−7.00, −2.55) | 70.38 (19.99) |
| | Sociodemographic adjusted | −10.39 (−13.06, −7.71) | −10.68 (−13.03, −8.32) | −14.71 (−17.61, −11.81) | |
| | Unadjusted | −11.07 (−13.94, −8.20) | −11.52 (−13.94, −9.10) | −15.98 (−18.95, −13.01) | |
| Vitality | Fully adjusted | −5.12 (−8.52, −1.72) | −4.00 (−7.40, −0.60) | −7.20 (−9.28, −5.12) | 63.97 (20.46) |
| | Sociodemographic adjusted | −12.58 (−16.06, −9.10) | −12.75 (−15.77, −9.73) | −18.67 (−21.48, −15.86) | |
| | Unadjusted | −13.23 (−16.83, −9.62) | −13.55 (−16.60, −10.49) | −19.86 (−22.72, −17.00) | |
| Social functioning | Fully adjusted | −7.53 (−10.67, −4.39) | −6.38 (−10.07, −2.69) | −10.68 (−13.48, −7.88) | 84.02 (23.53) |
| | Sociodemographic adjusted | −14.64 (−18.64, −10.63) | −14.67 (−18.10, −11.25) | −21.58 (−24.98, −18.18) | |
| | Unadjusted | −15.36 (−19.53, −11.20) | −15.58 (−19.00, −12.15) | −22.90 (−26.35, −19.45) | |
| Role emotional | Fully adjusted | −9.57 (−14.30, −4.84) | −8.12 (−13.44, −2.80) | −15.25 (−19.86, −10.62) | 84.79 (32.56) |
| | Sociodemographic adjusted | −17.66 (−23.28, −12.04) | −17.58 (−22.83, −12.34) | −27.47 (−32.97, −21.98) | |
| | Unadjusted | −18.53 (−24.40, −12.66) | −18.66 (−23.93, −13.39) | −29.03 (−34.68, −23.39) | |
| Mental health index | Fully adjusted | −6.50 (−9.58, −3.42) | −5.29 (−8.82, −1.76) | −9.30 (−11.36, −7.25) | 78.93 (16.14) |
| | Sociodemographic adjusted | −12.28 (−15.54, −9.03) | −12.22 (−15.26, −9.18) | −18.00 (−20.78, −15.23) | |
| | Unadjusted | −12.70 (−16.05, −9.35) | −12.71 (−15.78, −9.63) | −18.74 (−21.52, −15.95) | |

Reference group = 'Stable-low' loneliness.
Sociodemographic models were adjusted for age, gender, education, marital status, living arrangement, employment, area remoteness, country of birth and language spoken at home.
Fully adjusted models were adjusted for age, gender, education, marital status, living arrangement, employment, area remoteness, country of birth, language spoken at home, BMI, smoking, alcohol consumption, depression, anxiety, social support and stress.
*PCS* Physical Component Summary, *MCS* Mental Component Summary, *B* unstadardised Beta coefficients, *M* Mean, *SD* Standard deviation.

significantly associated with the overall physical health component, in addition to the mental health component and the sub-scales. A trajectory of 'Increasing' loneliness could be attributed to different life events such as illness, disability, retirement, caregiving responsibilities, and the loss of loved ones, which all contribute to a deterioration in physical health[30,60,65].

Our findings suggest that 'Increasing' loneliness, as well as 'Stable-medium', and 'Stable-high' loneliness classes, were all associated with current smoking, depression, anxiety, and stress. The association between loneliness and other risky health behaviours, such as smoking, highlights the broader health implications of loneliness. Loneliness has been an important and neglected population health determinant[66]. Individuals experiencing loneliness may turn to high-risk behaviours, such as smoking, as a coping mechanism or as a part of a broader pattern of unhealthy behaviours[67]. Similarly, the associations with depression, anxiety, and stress confirm that loneliness is intertwined with mental health[7,68,69]. There is evidence that loneliness may trigger a maladaptive biological stress response that can create further social withdrawal, which reinforces feelings of loneliness[70]. This cycle of loneliness and stress may contribute to the adoption of other maladaptive health behaviours, such as an unhealthy lifestyle, ultimately leading to adverse health outcomes[71].

## Limitations

Several limitations should be acknowledged. First, despite adjusting the analyses for demographics and health-related covariates, there may still be potential residual confounding arising from unmeasured or insufficiently measured confounders. In addition, there may be reverse causation bias due to underlying illnesses. However, by adjusting the models for baseline health-related quality of life, we minimised the risk for this potential bias. Second, the 3-year intervals at which data were collected raise the possibility that the reported loneliness levels during survey years may not precisely reflect the experience of loneliness in the interim periods. However, given the multi-wave nature of our study, it is reasonable to infer that the reported levels of loneliness during survey years offer a reasonably representative snapshot of the overall loneliness patterns. Third, self-reported data may not be accurate and are prone to biases. The one-item loneliness scale may be susceptible to underreporting due to societal stigma surrounding loneliness. In addition, there is no standardised cutoff point for the one-item loneliness measure. This ambiguity may have contributed to a variability in loneliness prevalence and its associated outcomes. However, the sensitivity analysis yielded consistent results across various cut points, further strengthening the robustness of our findings. In addition, the measures used in this study were found to have high validity and reliability in previous studies among Australian populations and longitudinal studies[43,72]. Despite these limitations, our findings still provide valuable insights into the association between loneliness trajectories and dimensions of health-related quality of life.

## Conclusions

Overall, our findings highlight the importance of social well-being for middle-aged and older women's physical and mental health. Our research reveals a strong longitudinal connection between elevated levels of loneliness over the course of 18 years and a range of subsequent health outcomes among middle-aged women. These findings suggest the importance of fostering social well-being as a universal public health approach in healthcare. There is a need to address loneliness, not only as a social concern but also as a determinant of both mental and physical well-being. Developing strategies to enhance social connections and alleviate loneliness can be considered in promoting overall health and improving health-related quality of life. Recognising the complex interplay between loneliness and health-related quality of life highlights the importance of holistic strategies that address the association between social, psychological, and physiological dimensions.

## Data availability

Data are publicly available and can be obtained upon formal permission from the ALSWH Data Access Committee at: https://alswh.org.au/for-data-users/applying-for-data/.

## Code availability

The code is available at: https://osf.io/wx8ay/.

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

## Acknowledgements

The research on which this study is based was conducted as part of the Australian Longitudinal Study on Women's Health by the University of Queensland and the University of Newcastle. We are grateful to the Australian Government Department of Health and Aged Care for funding and to the women who provided the survey data. D.D. is funded by the National Health and Medical Research Council Emerging Leader Fellowship (award No. 2009254). The funder had no role in study design, data collection and analysis, decision to publish or preparation of the manuscript.

## Author contributions

N.H.: design of the study, data analysis and writing of the manuscript. K.O. and P.J.C.: design of the study, data analysis and review and edit of the manuscript. D.M. and B.J.S.: design of the study and review and edit of the manuscript. D.D.: design of the study and writing of the manuscript.

## Competing interests

The authors declare no competing interests.
