## [Transparent Peer Review file · Communications Psychology]

Loneliness and long-term health-related quality of life: A latent class analysis among middle-aged and older Australian women

Corresponding Author: Dr Ding Ding

Version 0:

Decision Letter:

Dear Dr Ding,

Thank you for your patience during the peer-review process. Your manuscript titled "Loneliness and long-term health-related quality of life: A latent class analysis among middle-aged and older Australian women" has now been seen by 2 reviewers, and I include their comments at the end of this message. They find your work of interest but raised some important points. We are interested in the possibility of publishing your study in Communications Psychology, but would like to consider your responses to these concerns and assess a revised manuscript before we make a final decision on publication.

We therefore invite you to revise and resubmit your manuscript, along with a point-by-point response to the reviewers. Please highlight all changes in the manuscript text file.

Editorially, we consider it important that the revised manuscript provide additional robustness checks for the reported models and a justification for the categorization used to create binary loneliness.

I am attaching an Editorial Requests Table that details critical reporting requirements for the revised manuscript. Please attend to each item and ensure your manuscript is fully compliant. We are requesting that your manuscript aligns with these requirements as this facilitates the evaluation of your manuscript, reducing delays in re-review and potential future acceptance. If your revised manuscript is not aligned with these requests on major issues, such as those concerning statistics, it may be returned to you for further revisions without re-review. Additional information can be found in our style and formatting guide <https://www.nature.com/documents/commspsychol-style-formatting-guide-accept.pdf> Communications Psychology formatting guide.

Please use the following link to submit your

- revised manuscript,
- point-by-point response to the referees' comments,
- cover letter (as a separate document),
- the Editorial Policy Checklist (see below),
- the Reporting Summary (see below), and
- the completed Editorial Request Table (attached):

Link Redacted

We hope to receive your revised paper within 8 weeks; please let us know if you aren't able to submit it within this time so

that we can discuss how best to proceed. If we don't hear from you, and the revision process takes significantly longer, we may close your file. In this event, we will still be happy to reconsider your paper at a later date, provided it still presents a significant contribution to the literature at that stage.

Best regards,

Jennifer Bellingtier

Jennifer Bellingtier, PhD
Senior Editor
Communications Psychology

REVIEWER EXPERTISE:

Reviewer #1 Aging, Well-Being

Reviewer #2 Health Psychology

REVIEWER REPORTS:

Reviewer #1 (Remarks to the Author):

This is an interesting study investigating the association between long-term patterns of loneliness and health-related quality of life (HRQoL) among middle-aged and older Australian women. The study utilizes data from the Australian Longitudinal Study of Women's Health (ALSWH), spanning 18 years, and applies latent class analysis to identify distinct loneliness trajectories. There are a lot of strengths in this study. The longitudinal nature, the large sample size, the careful use of latent class analysis with BCH estimation method to accommodate the uncertainty in classification, the compliance with STROBE guidelines, the use of multiple imputation for missing data, and the careful interpretation of the findings and acknowledgement of limitations. I agree that the study will contribute well to the literature. I only have a few minor comments to improve the manuscript further:

1. I understand the constraint regarding data sharing in this type of dataset. However, I would like to encourage the authors to at least share their analytic code to ensure the reproducibility of the findings.
2. The reliability or internal consistency of the measures should be reported in the Measures section
3. While the latent class analysis and subsequent regression models are appropriate, the paper would benefit from additional robustness checks, such as sensitivity analyses, to assess the stability of the identified classes and the consistency of the regression results across different model specifications

Reviewer #2 (Remarks to the Author):

This manuscript describes an analysis of longitudinal data collected over 18 years from a national sample of Australian women. The aim of the research was to identify common patterns of loneliness over time, and relations between loneliness trajectories and HRQoL. The manuscript is extremely well written, and the work appears to have been thoughtfully planned and executed. The work makes a novel and interesting contribution to the evidence base in relation to loneliness. The application of a person-centred approach, in the form of a latent class analysis was appropriate to address the aims of the research. As such, I commend the authors, and have some very minor suggestions to make.

1. There is a typo on line 202 ('The' at the beginning of the sentence is not capitalized).
2. The decision to categorise the binary loneliness variable at the mid point of the response scale seems reasonable - however, I'm interested to know if there was any specific rationale that informed that decision, in favour of an alternate split (e.g., rarely/none v any loneliness)? Patterns of loneliness may differ if the variable was dichotomized differently - so it would be appropriate to list this in the limitations section.

Kind regards,

Jen Olson, PhD
Senior Methodologist
Ottawa Hospital Research Institute

EDITORIAL POLICIES

We ask that you ensure your manuscript complies with our editorial policies and reporting requirements.

To that end, we require revised manuscripts to be accompanied by two completed items: a reporting summary that collects information on study design and procedure, and an editorial policy checklist that verifies compliance with all required editorial policies.

- <https://www.nature.com/documents/nr-reporting-summary.zip>>Nature Research Reporting Summary
- <https://www.nature.com/documents/nr-editorial-policy-checklist.pdf>>Editorial Policy Checklist

All points on the policy checklist must be addressed. Your revised manuscript can only be sent back to the referees if these checklists are completed and uploaded with the revision.

Notes: If you have submitted a Stage 1 Registered Report, Review, Primer, Comment, or Perspective you do not need to submit these forms. If you have already submitted these forms, you may disregard this request.

** Visit Nature Research's author and referees' website at <http://www.nature.com/authors>>www.nature.com/authors for information about policies, services and author benefits**

If you experience problems in linking your ORCID, please contact the <http://platformsupport.nature.com/>>Platform Support Helpdesk.

Version 1:

Decision Letter:

Dear Dr Ding,

Your manuscript titled "Loneliness and long-term health-related quality of life: A latent class analysis among middle-aged and older Australian women" has now been seen by our reviewers, whose comments appear below. In light of their advice I am delighted to say that we are happy, in principle, to publish a suitably revised version in Communications Psychology.

We therefore invite you to revise your paper one last time to address the remaining concerns of our reviewers and a list of editorial requests. At the same time we ask that you edit your manuscript to comply with our format requirements and to maximise the accessibility and therefore the impact of your work.

EDITORIAL REQUESTS:

SUBMISSION INFORMATION:

In order to accept your paper, we require the files available at <https://www.nature.com/documents/commsj-file-checklist.pdf>.

OPEN ACCESS:

* DATA AVAILABILITY:

Link Redacted

Best regards,

Jennifer Bellingtier

Jennifer Bellingtier, PhD
Senior Editor
Communications Psychology

REVIEWERS' EXPERTISE:

Reviewer #1 Aging, Well-Being
Reviewer #2 Health Psychology

REVIEWERS' COMMENTS:

Reviewer #1 (Remarks to the Author):

The authors have addressed all my comments thoroughly, and I sincerely appreciate their efforts. I have only one minor suggestion: I recommend uploading the codes to a repository such as OSF or ResearchBox, rather than GitHub. This will enhance transparency, promote reproducibility, and provide easier access for readers and future researchers.

Reviewer #2 (Remarks to the Author):

Thank you for the opportunity to review the revised version of this manuscript. I am satisfied that the revisions address my comments made in the original review. This is an interesting manuscript and from my perspective, is suitable for publication in its current form.

Dear reviewers,

We thank you for your constructive feedback and the opportunity to revise and resubmit the manuscript. We have revised our manuscript according to the comments and believe that these comments have helped us improve this paper. We provide a point-by-point response to these comments below.

On behalf of all the authors,

Kind regards,

Neta Hagani, Ding (Melody) Ding

Response to reviewers:

Reviewer #1 (Remarks to the Author):

This is an interesting study investigating the association between long-term patterns of loneliness and health-related quality of life (HRQoL) among middle-aged and older Australian women. The study utilizes data from the Australian Longitudinal Study of Women's Health (ALSWH), spanning 18 years, and applies latent class analysis to identify distinct loneliness trajectories. There are a lot of strengths in this study. The longitudinal nature, the large sample size, the careful use of latent class analysis with BCH estimation method to accommodate the uncertainty in classification, the compliance with STROBE guidelines, the use of multiple imputation for missing data, and the careful interpretation of the findings and acknowledgement of limitations. I agree that the study will contribute well to the literature. I only have a few minor comments to improve the manuscript further:

1. I understand the constraint regarding data sharing in this type of dataset. However, I would like to encourage the authors to at least share their analytic code to ensure the reproducibility of the findings.

Response: Thank you for your comments. We agree with the comment and provide a link to the code that was used (p. 17, lines 389-390):

Code availability: the code is available at: <https://github.com/Netahag/Loneliness-and-long-term-health-related-quality-of-life.git>

2. The reliability or internal consistency of the measures should be reported in the Measures section

Response: We have added the internal consistency of the measures on p. 6, lines 117-121:

One-item loneliness measures were previously shown to be reliable ($r_{xx} > 0.70$, [37]) and valid (strongly correlated with other more comprehensive, common loneliness measures such as the UCLA and the de Jong loneliness scales, as well as with the indirect one-item loneliness measures [37-39]).

And on p. 6-7, lines 138-143:

The SF-36 is a validated and reliable measure that was examined among a variety of populations, including middle-aged and older adults [41, 42]. In a population-representative Australian household survey (Australian National Household Survey), the internal consistency ranged between 0.82-0.93 across the HRQoL dimension [41]. Similar reliability was observed in other studies among diverse populations [43-45].

3. While the latent class analysis and subsequent regression models are appropriate, the paper would benefit from additional robustness checks, such as sensitivity analyses, to assess the stability of the identified classes and the consistency of the regression results across different model specifications

Response: Thank you for your comment. We have added two sensitivity analyses to assess the stability of the identified classes and the consistency of the regression results on p. 13, lines 285-298 and Appendix 4, Tables S4.1-S4.5:

Sensitivity analysis

Results from the probit model were consistent with the main analyses (**Tables S4.1-S4.2**). When using the different loneliness cut-points, we found that the four-class model was still the optimal fit (**Table S4.3**) and similar associations were found between the predictors of the latent class membership and the four classes. However, as expected when defining the loneliness group as those who responded as being lonely “some or a little of the time”, the associations became weaker and less precise than those in the main analysis, except in the cases of depression and social support (**Table S4.4**). Attenuations were also in the associations between loneliness classes and HRQoL components in the unadjusted, sociodemographic-adjusted and fully adjusted models. There were no significant associations in the ‘Increasing’ loneliness class compared to the ‘Stable low’. However, in the ‘Stable medium’ and ‘Stable high’ classes, loneliness was significantly associated with almost all of the HRQoL components (**Table S4.5**). Overall, these sensitivity analyses have demonstrated the robustness of our findings.

Reviewer #2 (Remarks to the Author):

This manuscript describes an analysis of longitudinal data collected over 18 years from a national sample of Australian women. The aim of the research was to identify common

patterns of loneliness over time, and relations between loneliness trajectories and HRQoL. The manuscript is extremely well written, and the work appears to have been thoughtfully planned and executed. The work makes a novel and interesting contribution to the evidence base in relation to loneliness. The application of a person-centred approach, in the form of a latent class analysis was appropriate to address the aims of the research. As such, I commend the authors and have some very minor suggestions to make.

1. There is a typo on line 202 ('The' at the beginning of the sentence is not capitalized).

Response: Thank you for your comment. We revised the sentence on p. 10, lines 225-226: The mean loneliness across waves 3-8 among individuals in class 1 was 33.5%.

2. The decision to categorise the binary loneliness variable at the mid point of the response scale seems reasonable - however, I'm interested to know if there was any specific rationale that informed that decision, in favour of an alternate split (e.g., rarely/none v any loneliness)? Patterns of loneliness may differ if the variable was dichotomized differently - so it would be appropriate to list this in the limitations section.

Response: We added information on the loneliness cutoff point to the limitation section on p. 16, lines 365-368. We have also conducted a series of sensitivity analyses which have demonstrated the robustness of our findings based on the current categorisation (Appendix 4, Tables S4.1-S4.5).

In addition, there is no standardised cutoff point for the one-item loneliness measure. This ambiguity may have contributed to a variability in loneliness prevalence and its associated outcomes. However, the sensitivity analysis yielded consistent results across various cut points, further strengthening the robustness of our findings.